# Molecular characterization of CHAD domains as inorganic polyphosphate-binding modules

Laura Lorenzo-Orts, Ulrich Hohmann, Jinsheng Zhu, Michael Hothorn

Inorganic polyphosphates (polyPs) are linear polymers of orthophosphate units linked by phosphoanhydride bonds. Here, we report that bacterial, archaeal, and eukaryotic conserved histidine α-helical (CHAD) domains are specific polyP-binding modules. Crystal structures reveal that CHAD domains are formed by two four-helix bundles, giving rise to a central pore surrounded by conserved basic surface patches. Different CHAD domains bind polyPs with dissociation constants ranging from the nano- to mid-micromolar range, but not nucleic acids. A CHAD—polyP complex structure reveals the phosphate polymer binding across the central pore and along the two basic patches. Mutational analysis of CHAD—polyP interface residues validates the complex structure. The presence of a CHAD domain in the polyPase ygiF enhances its enzymatic activity. The only known CHAD protein from the plant *Ricinus communis* localizes to the nucleus/nucleolus when expressed in Arabidopsis and tobacco, suggesting that plants may harbor polyPs in these compartments. We propose that CHAD domains may be used to engineer the properties of polyP-metabolizing enzymes and to specifically localize polyP stores in eukaryotic cells and tissues.

## Introduction

Inorganic polyphosphates (polyPs) form an important phosphate ($P_i$) and energy store in pro- and eukaryotic cells (Brown & Kornberg, 2004; Rao et al, 2009). In bacteria, polyPs can form granules in the nucleoid region, regulate the cell cycle (Racki et al, 2017), form cation-selective membrane channels (Pavlov et al, 2005), control cell motility (Rashid & Kornberg, 2000), and mediate cellular stress responses, for instance, by preventing protein aggregation (Gray et al, 2014). In eukaryotes, polyPs have thus far been found in vacuoles or specialized acidocalcisomes (Lander et al, 2016) and form an important store for $P_i$ (Ogawa et al, 2000; Hothorn et al, 2009; Gerasimaitė et al, 2014; Desfougères et al, 2016) and divalent metal ions (Docampo & Huang, 2016; Klompmaker

et al, 2017). At the physiological level, polyPs are involved in cell cycle control (Bru et al, 2016), cell death responses (Abramov et al, 2007), blood coagulation (Müller et al, 2009), skeletal mineralization (Omelon et al, 2009), and in the post-translational modification of proteins (Azevedo et al, 2015).

PolyP-metabolizing enzymes have been well characterized in bacteria and lower eukaryotes. In bacteria, polyP may be synthesized from ATP by the polyphosphate kinase 1 (PPK1) (Kornberg, 1957; Ahn & Kornberg, 1990) or from ATP/GTP by PPK2 (Ishige et al, 2002; Zhang et al, 2002; Nocek et al, 2008; Parnell et al, 2018). In lower eukaryotes such as fungi, protozoa, and algae, polyP is generated from ATP by the membrane-integral Vacuolar Transporter Chaperone (VTC) complex (Boyce et al, 2006; Hothorn et al, 2009; Aksoy et al, 2014; Kohl et al, 2018). No polyphosphate kinase has been reported from higher eukaryotes thus far, despite the presence of polyPs in these organisms (Kumble & Kornberg, 1995). Exopolyphosphatase PPX1 (Akiyama et al, 1993) and the triphosphate tunnel metalloenzyme (TTM) ygiF (Martinez et al, 2015) are polyP-degrading enzymes in bacteria. Eukaryotic polyphosphatases include the yeast exopolyphosphatase 1 (PPX1) (Wurst & Kornberg, 1994), the endopolyphosphatases PPN1 (Kumble & Kornberg, 1996) and PPN2 (Gerasimaitė & Mayer, 2017), the Ddp1-type Nudix hydrolases (Lonetti et al, 2011), human H-prune (Tammenkoski et al, 2008), and the plant tripolyphosphatase TTM3 (Moeder et al, 2013; Martinez et al, 2015).

To date, no polyP-binding domain has been identified, although an engineered polyP-binding domain from EcPPX1 has been used to immunolocalize polyPs in eukaryotic cells and tissues (Werner et al, 2007b). We have previously identified a small, helical domain at the C-terminus of the bacterial short-chain polyphosphatase ygiF (Kohn et al, 2012; Martinez et al, 2015). This domain of unknown function has been annotated as CHAD (conserved histidine α-helical domain, PFAM PF05235) (Iyer & Aravind, 2002). Many CHAD domain–containing proteins harbor an N-terminal TTM domain, whereas stand-alone CHAD proteins are often part of operons expressing polyP-metabolizing enzymes (Iyer & Aravind, 2002). Recently, it was found that CHAD domain–containing proteins specifically localize to polyP granules in the bacterium *Ralstonia eutropha* (Tumlirsch & Jendrossek, 2017).

Structural Plant Biology Laboratory, Department of Botany and Plant Biology, Faculty of Sciences, University of Geneva, Geneva, Switzerland

Correspondence: michael.hothorn@unige.ch
Ulrich Hohmann's present address is Institute of Molecular Biotechnology of the Austrian Academy of Sciences and Research Institute of Molecular Pathology, Vienna Biocenter, Vienna, Austria

In this study, we characterize CHAD domains as bona fide polyP-binding modules.

# Results

We located CHAD domains in the different kingdoms of life. According to Interpro (https://www.ebi.ac.uk/interpro), ~99% of the annotated CHAD proteins correspond to bacteria, whereas only ~1% (129 proteins) and 0.1% (10 proteins) belong to archaea and eukaryota, respectively (Fig 1A). We selected CHAD domain–containing proteins belonging to the three kingdoms of life: archaea (*Sulfolobus solfataricus*; termed SsCHAD hereafter), bacteria (*Chlorobium tepidum*; CtCHAD), and eukaryota (*Ricinus communis*

or castor bean; RcCHAD) (Figs 1A and S1). Several of these CHAD proteins form part of gene clusters encoding polyP-metabolizing enzymes, with the exception of RcCHAD (Fig 1B).

To confirm if indeed RcCHAD is expressed in *Ricinus*, we performed reverse transcription PCR (RT-PCR) experiments using *Ricinus* cDNA prepared from leaf extracts. We detected a transcript corresponding to the predicted *RcCHAD* sequence (Figs 1C and S2). We next expressed RcCHAD carrying a C-terminal mCherry tag under the control of a constitutive promoter in the model plant *Arabidopsis thaliana*. We found that the fusion protein specifically localized to the nucleus and nucleolus of hypocotyl and root cells (Fig 1D).

We next sought to determine crystal structures for different CHAD domains. Diffraction-quality crystals developed for RcCHAD

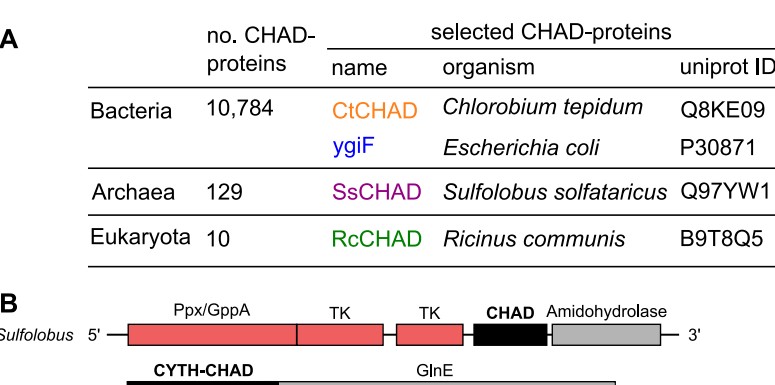

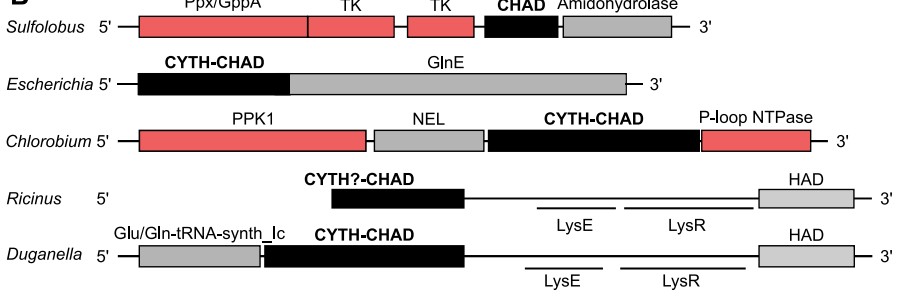

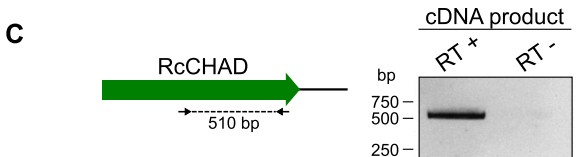

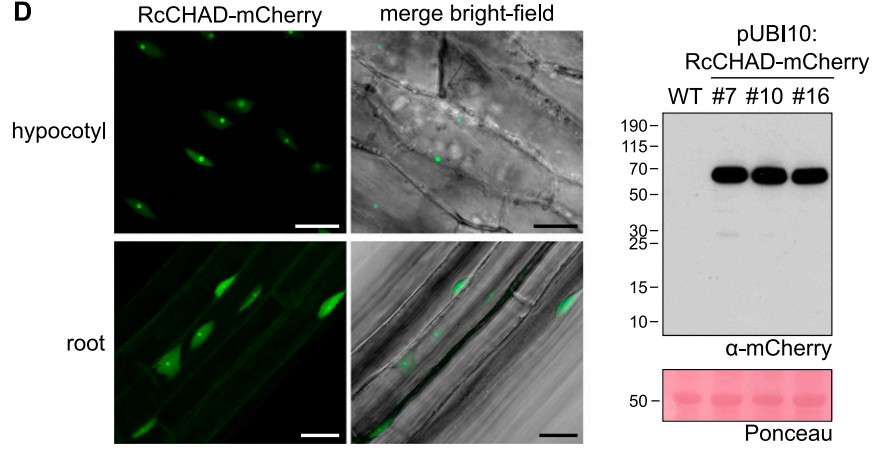

**Figure 1. CHAD domain–containing proteins are present in all kingdoms of life.**
**(A)** Overview of CHAD domain–containing proteins characterized in this study. **(B)** Genetic loci of genes encoding CHAD domain–containing proteins (in black). Genes coding for polyP-metabolizing enzymes are highlighted in red. The DNA sequence upstream of *RcCHAD* is missing from contig RCOM_0386220 in the NCBI database. **(C)** RT-PCR using primers binding to the *RcCHAD* coding sequence (left) results in a specific product amplified from cDNA from *R. communis* leaves (right; DNA sequence in Fig S2). **(D)** Confocal microscopy of transgenic Arabidopsis lines expressing Ubi10p: RcCHAD-mCherry reveals RcCHAD to localize in the nucleus and nucleolus of root and hypocotyl cells (left, scale bars correspond to 20 μm). A Western blot using an anti-mCherry antibody reveals a specific band migrating at the predicted size of the RcCHAD-mCherry fusion protein (63 kD). The Ponceau-stained membrane is shown as loading control below (the major 55-kD band corresponds to RuBisCo).

and CtCHAD, whereas crystals of SsCHAD diffracted only to ~7 Å. Initial attempts to determine the RcCHAD structure using the molecular replacement method and the isolated CHAD domain of ygiF[221–422] or an unpublished CtCHAD structure (Protein Data Bank [PDB] ID 3E0S; both sharing ~30% sequence identity with RcCHAD) failed (see the Material and Methods section). We, thus, used the moderate anomalous signal present in the native RcCHAD dataset to locate a single $Zn^{2+}$ ion. The structure was solved using the single-wavelength anomalous dispersion method, and the refined model revealed a $Zn^{2+}$ ion tetrahedrally coordinated by His50 and His136 originating from two symmetry-related RcCHAD molecules (Fig S3). It has been previously speculated that CHAD domains may bind divalent cations using conserved histidine residues (Iyer & Aravind, 2002). We found, however, that His50 and His136 from the RcCHAD $Zn^{2+}$-binding site are not conserved among other CHAD domains (Fig S1), and consistently no metal ion–binding sites were found in our crystal structures of CtCHAD (root mean square deviation [r.m.s.d.] to the deposited PDB-ID 3E0S is ~0.6 Å comparing 290 corresponding $C_{\alpha}$ atoms, Fig S4) or ygiF (Martinez et al, 2015).

We next compared the refined structures of the plant CHAD domain to the bacterial CtCHAD and ygiF structures. All CHAD domains fold into two 4-helix bundles with up-down-up-down topology (Figs 2A and S4). In the available structures, the helical

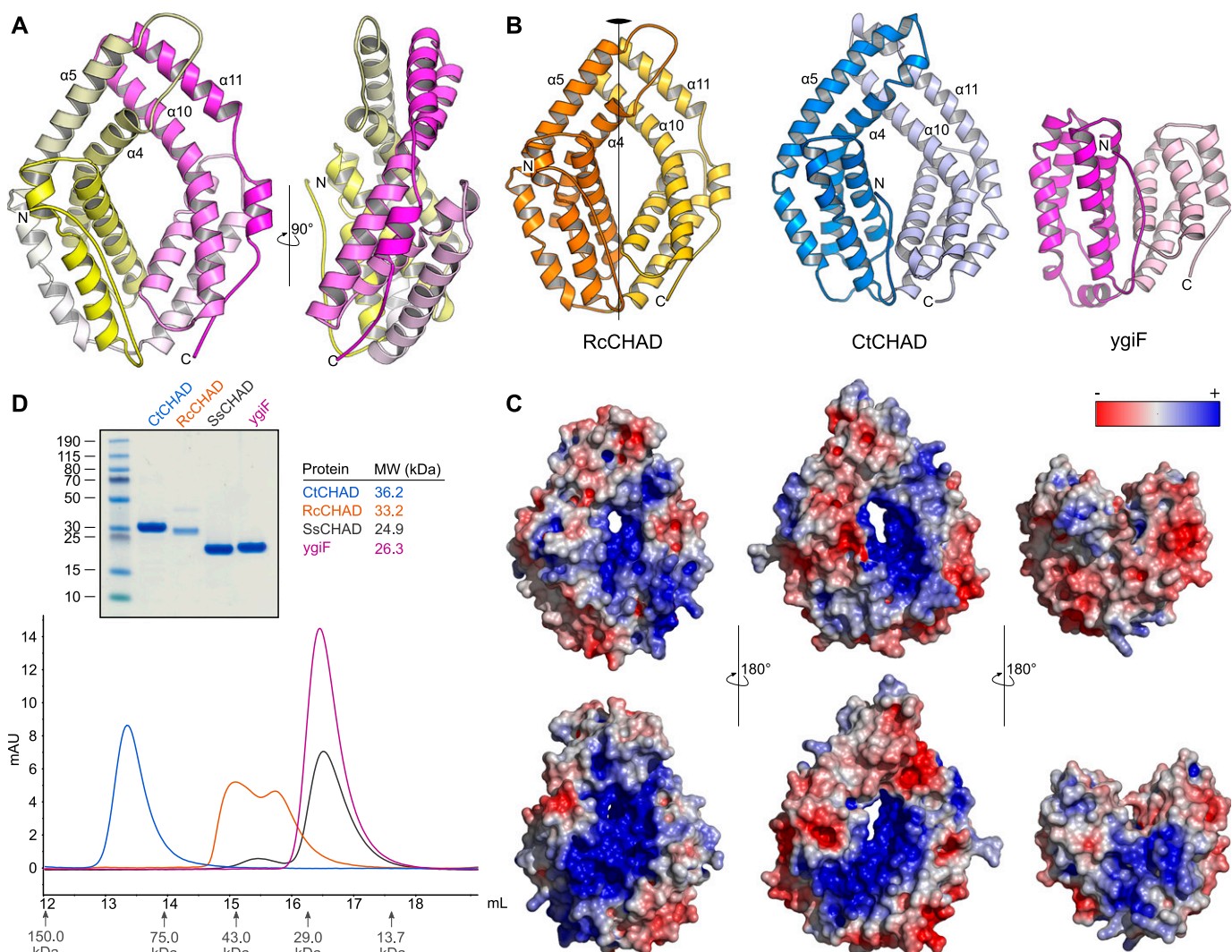

**Figure 2. CHAD domains are helical bundles with twofold internal symmetry and a conserved basic surface area.**
**(A)** Architecture of the CHAD domain. Shown is a ribbon diagram of RcCHAD colored from N- (yellow) to C-terminus (magenta). **(B)** Structural comparison of the RcCHAD, CtCHAD, and ygiF CHAD domain (Martinez et al, 2015) structures reveal the presence of two 4-helix bundles in all CHAD domains, related by pseudo twofold symmetry (indicated by a vertical line). Note that the helices α4, α6, α10, and α11 in ygiF are much shorter when compared with RcCHAD and CtCHAD, and hence ygiF lacks the central pore. **(C)** Identification of a conserved basic surface area in pro- and eukaryotic CHAD domains. Electrostatic potentials calculated in APBS (Jurrus et al, 2018) were mapped onto CHAD domain molecular surfaces in Pymol. Shown are front (upper panel) and back (lower panel) views. A highly basic surface area covers the front- and back side of the CHAD domain and includes the central pore present in RcCHAD and CtCHAD. **(D)** Analytical size-exclusion chromatography reveals different oligomeric states for the CHAD domains analyzed in this study. An SDS–PAGE analysis of the respective peak fractions (pooled) is shown alongside.

bundles are related by an almost perfect twofold axis and can be superimposed with r.m.s.d.'s ranging from 2.2 to 2.7 Å (Fig S4). Notably, helices α4/α5 and α10/α11 are protruding the bundle cores in the RcCHAD and CtCHAD structures, forming a small pore in the center of the domain, which we find to be absent in our ygiF structure (Figs 2B and S4). This rationalizes the presence of "long" (~300 amino acids, e.g., RcCHAD and CtCHAD) and "short" (~200 amino acids, e.g., ygiF and SsCHAD) CHAD domains. Analysis of the surface charge distribution in the different CHAD domains revealed a highly basic central cavity, which is surrounded by two basic surface patches on each side (Fig 2C). The basic amino acids contributing to these surface patches are highly conserved among different CHAD proteins (Fig S1). Similarly, highly basic surface patches are present in many polyP-metabolizing enzymes (Fig S5). Analytical size-exclusion chromatography experiments indicated that the different CHAD proteins adopt different oligomeric states in solution, with ygiF and SsCHAD behaving as monomers, whereas unliganded CtCHAD form dimers in solution (Figs 2D and S8, see the Discussion section).

Given their highly basic surface charge distribution, the fact that CHAD domains are found in polyP-metabolizing enzymes or gene clusters (Iyer & Aravind, 2002) and that they can localize to polyP bodies (Tumlirsch & Jendrossek, 2017), we next tested if CHAD domains directly bind polyPs.

We assayed polyP-binding of RcCHAD, SsCHAD, CtCHAD, and ygiF in quantitative grating-coupled interferometry (GCI) experiments (see the Material and Methods section). Biotinylated polyP (chain length ~100 $P_i$ units) was coupled to the GCI chip, and different proteins were used as analytes (Fig S6). For our different CHAD domains, dissociation constants ($K_D$) for polyP cover the nanomolar to the mid-micromolar range (Fig 3A). The yeast polyP polymerase Vtc4p was used as a positive control (Hothorn et al, 2009) and BSA as negative control (Fig 3A). RcCHAD and SsCHAD bind polyP with a one-to-one kinetics and with a $K_D$ of 6 μM and 45 nM, respectively (Fig 3A). In the case of CtCHAD and ygiF, the sensograms could not be explained by a simple one-to-one binding model. Instead, we observed two distinct association and dissociation events. A heterogeneous analyte model was used to fit the data (CtCHAD $K_D^1$ = 1.9 μM, $K_D^2$ = 147 nM; ygiF $K_D^1$ = 2.2 μM, $K_D^2$ = 40.5 μM). We performed competition experiments adding polyP (average chain length ~7 $P_i$ units) in various concentrations to a fixed concentration of CHAD protein sample used as analyte (Fig 3B). In agreement with our direct binding assay, we find that polyP can efficiently compete for binding of SsCHAD and RcCHAD to the polyP-labeled surface of the GCI chip, with estimated $IC_{50}$'s of ~1 μM (Fig 3B). In contrast, the highly negatively charged diadenosine pentaphosphate (AP5A) did not efficiently compete for binding of RcCHAD to polyP in GCI assays (Fig 3C), and we could not observe detectable binding of SsCHAD or RcCHAD to GCI chips coated with biotinylated single-stranded DNA or RNA (Fig 3D). However, heparin, which is absent in bacteria and plants, was bound by CtCHAD with binding constants comparable to those determined for polyP (Fig 3D). Together, our quantitative binding assays suggest that CHAD domains interact with polyPs with specificity and selectivity. In line with this, we found that the presence of a C-terminal CHAD domain stimulated the previously reported tripolyphosphatase activity of ygiF (Fig 3E) (Martinez et al, 2015).

Next, we sought to identify residues in the CHAD domain involved in polyP coordination. A 1.7 Å structure of CtCHAD derived from crystals grown in 2 M $(NH_4)_2SO_4$ revealed five sulfate ions bound in the central basic cavity of the CHAD domain (Fig 4A). Crystals of CtCHAD grown in the presence of polyP (average chain length ~7 $P_i$ units) diffracted to 2.1 Å resolution and revealed continuous electron density transversing the central pore. The refined model includes a polyP 9-mer bound in the center of the CHAD domain and a tripolyphosphate moiety located at the distal side of the second basic surface patch (Fig 4B). Additional peaks in the difference electron density map were too weak to be interpreted (dashed line in Fig 4B). Superposition of the refined sulfate and polyP-bound CtCHAD structures (r.m.s.d. is ~0.5 Å comparing 304 corresponding $C_α$ atoms) suggests that the sulfate ions mimic the positions of $P_i$ units in the polyP chain binding across the CHAD domain center (Fig 4C). The overall mode of polyP binding in CtCHAD is similar to the one seen in the previously reported Vtc4—polyP and PPK2—polyP complex structures (Hothorn et al, 2009; Parnell et al, 2018), with the polymer binding along a highly basic, solvent-exposed surface (Figs 4B and S5).

We validated our structural model by mutational analysis of polyP-interacting residues. In the CtCHAD—polyP complex structure, an apparent polyP 9-mer is coordinated by a set of conserved lysine and arginine residues lining the central cavity (Figs 5A and S1). We mutated His253, Arg256, and Arg260, which form a hydrogen-bonding network with polyP (Fig 5A), to alanine. The mutant proteins bind polyP with sixfold to eightfold reduced affinity (Fig 5B). Mutation of the corresponding residues His29, Arg32, and Arg36 in SsCHAD to alanine resulted in an ~25-fold reduction in binding (Figs 3A, 5B, and S7). Additional mutation of Arg296 and Arg418/Tyr419 to alanine in CtCHAD led to ~80-fold to 150-fold reduction in binding when compared with wild-type CtCHAD, whereas a His29/Arg32/Arg36/Arg69 SsCHAD quadruple mutant shows no detectable polyP binding in our GCI assay (Figs 3A, 5A, and B, and S7). Together, these experiments reveal that the conserved basic amino acids surrounding the central cavity in different CHAD proteins are involved in the specific recognition of phosphate polymers.

Given that our analysis of different CHAD domain structures revealed the presence of additional, large, and conserved basic surface patches, we next asked if these surfaces may be involved in the binding of long polyP chains. To this end, we generated additional point mutations in CtCHAD targeting either the "front" or the "back" side of the domain (shown in cyan and yellow in Fig 5C, respectively). Mutation of the conserved Arg438 and Lys441 to alanine resulted in an ~2-fold to 10-fold reduced binding affinity (Figs 5C and D, and S1). Mutation of Arg385, Lys386, Lys389, and Lys390 on the "back" side of the domain had a similar effect (Fig 5C and D). In line with this, longer polyPs (~30 $P_i$ units) compete much more efficiently ($IC_{50}$ ~ 20 nM) with CtCHAD binding to the polyP-coated (~100 $P_i$ units) GCI chip, when compared with polyP 7-mers ($IC_{50}$ ~ 0.6 μM) or tripolyphosphate ($IC_{50}$ ~ 0.2 μM) (Figs 5E and S6). Together, these experiments suggest that CHAD domains can bind long polyP chains using their entire basic surface patch covering the "front" and "back" sides of the domain, as well as the central pore.

Our finding that CHAD domains can specifically bind polyPs with high affinity prompted us to further dissect the nuclear/nucleolar

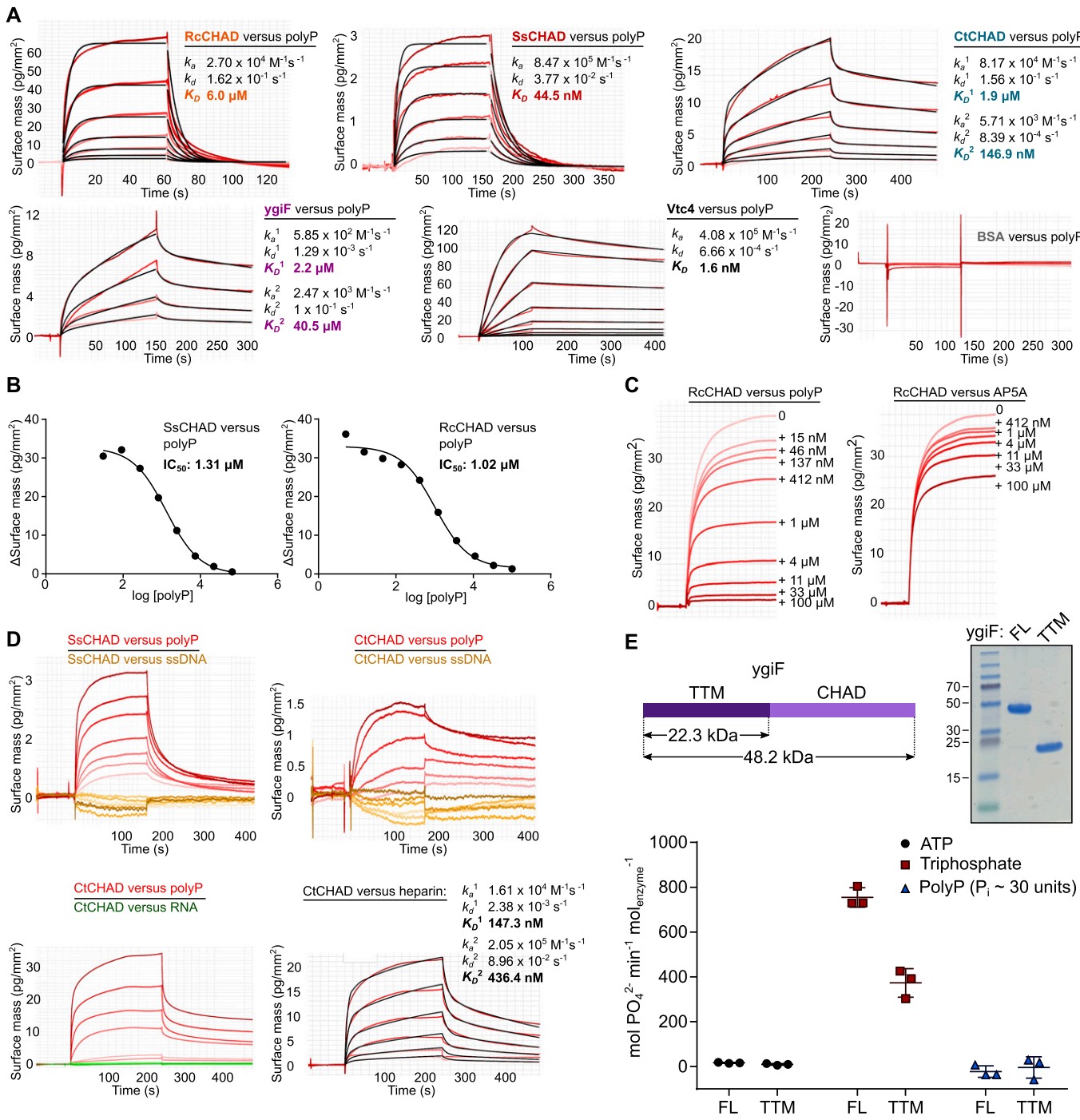

**Figure 3. CHAD domains are specific polyP-binding modules.**
**(A)** Quantitative GCI polyP-binding assay. Biotinylated polyP (average chain length is ~100 Pi units) was immobilized on a streptavidin GCI chip, and the different CHAD domains were used as analytes (Fig S6). The yeast polyP polymerase Vtc4 and BSA were used as positive and negative controls, respectively. Shown are recorded sensograms (in red) with the respective fits (in black) and including table summaries of the derived association rate constant ($k_a$), dissociation rate constant ($k_d$), and dissociation constant ($K_D$). **(B)** Short-chain polyPs (average chain length ~7 Pi units) can compete with CHAD domains for binding to the polyP-coated GCI chip. Shown are dose–response curves with derived $IC_{50}$ estimates. **(C)** Diadenosine pentaphosphate (AP5A) cannot compete with RcCHAD for binding to the polyP-coated GCI chip (right panel) as efficiently as polyP (left panel, average chain length ~7 Pi units). Shown are sensograms of the association phase at indicated inhibitor concentration. **(D)** Sensograms for SsCHAD and RcCHAD reveal no significant interaction with biotinylated single-stranded DNA (54 nt, in orange) or single-stranded RNA (10 nt, in green). Biotinylated polyP (average chain length ~100 Pi units, in red) is shown as a positive control. CtCHAD, however, binds biotinylated heparin. Shown are the recorded sensograms (in red) with the respective fits (in black) and including a table summary of the derived association rate constant ($k_a$), dissociation rate constant ($k_d$), and dissociation constant ($K_D$). **(E)** Phosphohydrolase activities of ygiF full-length 1–433 (FL) and ygif-TTM$^{1–200}$ (TTM) versus different phosphorylated substrates. Symbols represent raw data, lines indicate mean values, and error bars denote SD of three independent replicates. An SDS–PAGE analysis of the purified proteins is shown alongside. The theoretical molecular weight is ~22.3 kD for TTM and ~48.4 kD for FL.

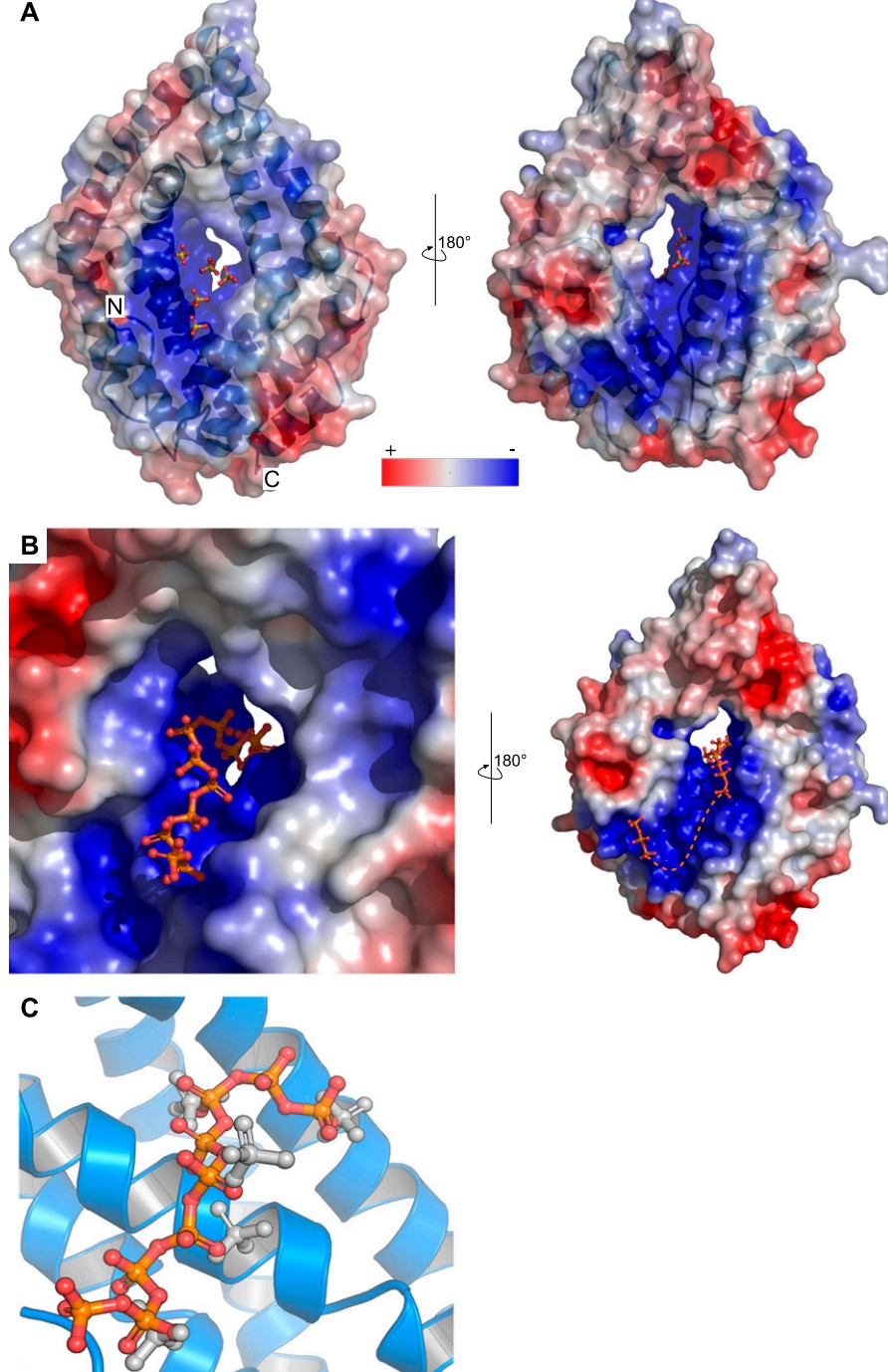

**Figure 4. The basic surface area in CHAD domains provides a binding platform for polyPs.**
**(A)** Sulfate ions (in bonds representation) originating from the crystallization buffer are bound to the basic surface area in CtCHAD. Shown are front and back views of CtCHAD as combined ribbon diagram and molecular surface. An electrostatic potential calculated in APBS was mapped onto the molecular surfaces in Pymol. **(B)** Overview of the polyP complex structure, obtained by crystallization of CtCHAD in the presence of 5 mM polyP (average length ~7 $P_i$ units). A polyP 9-mer and a tripolyphosphate moiety could be modeled (in bonds representation), with the polyP 9-mer occupying the central pore and extending to both sides. The dashed line indicates the approximate position of several peaks in the $F_o$-$F_c$ difference electron density map, which could not be modeled with confidence. **(C)** Structural superposition of the sulfate ion– and polyP-bound CtCHAD structures (r.m.s.d. is ~0.5 Å comparing 304 corresponding $C_\alpha$ atoms) reveals that the sulfate ions (in bonds representation, in gray) mimic the position of Pi units in the polyP 9-mer (in orange-red) in the CtCHAD-polyP complex.

localization of RcCHAD stably expressed in Arabidopsis (Fig 1D). It is presently unknown if polyPs are present in plants and where they would be localized. We transiently expressed RcCHAD-mCherry in tobacco leaves and again found the fusion protein to localize to the nucleus and to be further enriched in the nucleolus (Fig 6A–D). Next, we co-expressed RcCHAD-mCherry together with the bacterial polyP kinase EcPPK1. Expression of EcPPK1 has been previously reported to lead to cytosolic polyP accumulation in yeast cells (Gerasimaitė et al, 2014). Notably, RcCHAD re-localized to punctuate structures in the cytosol, which we assume to represent EcPPK1-generated polyP bodies (Fig 6A and D). Consistently, co-expression of RcCHAD-mCherry with a catalytically inactive variant of EcPPK1-mCitrine did not affect the nuclear/nucleolar localization of the CHAD domain in tobacco (Fig 6A and D). Based on these experiments, we speculate that RcCHAD may bind to a nucleolar/nuclear polyP pool in tobacco and in

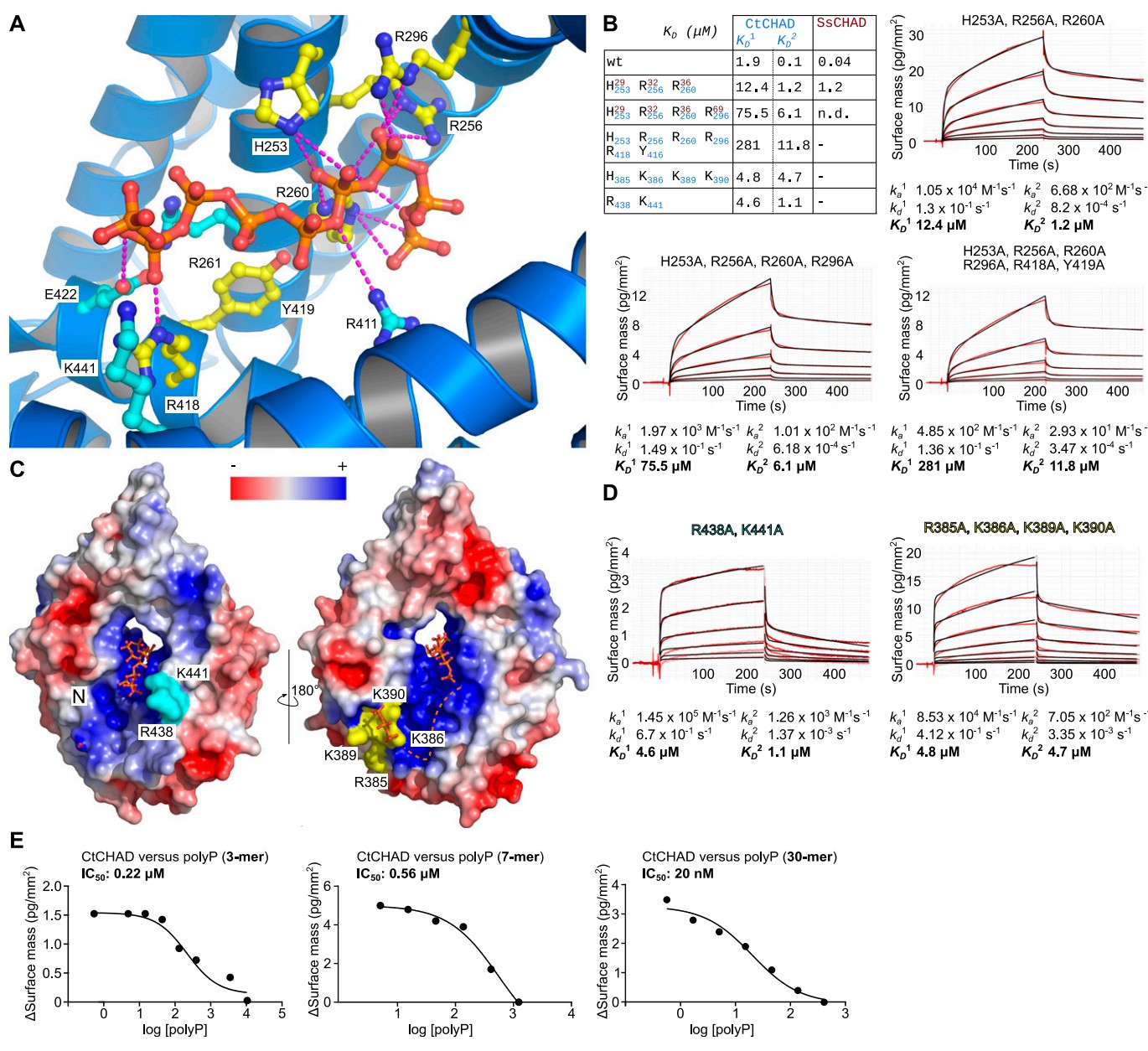

**Figure 5. CtCHAD binds polyP through basic amino-acid residues distributed along the central cavity and the back of the protein.**
**(A)** Detailed view of CtCHAD (blue ribbon diagram) bound to a polyP 9-mer (in orange, in bonds representation) and including selected conserved basic amino acids involved in polyP binding (in cyan, residues included in mutational analyses shown in yellow). **(B)** Mutations in the central basic binding surface in CtCHAD (dissociation constants in blue; corresponding mutations in SsCHAD in red) strongly decrease polyP binding in GCI assays. Shown are sensograms (in red), the respective fits (in black), and table summaries of the derived association rate constant ($k_a$), dissociation rate constant ($k_d$), and dissociation constant ($K_D$). **(C)** Identification of two distal polyP-binding surfaces on the front side (shown in cyan) and back side (in yellow) of the CtCHAD—polyP complex structure. **(D)** Mutations of conserved residues in the two distal surfaces reduce polyP binding as shown in GCI assays. Shown are sensograms (in red), the respective fits (in black), and table summaries of the derived kinetic parameters. **(E)** GCI competition assays using tripolyphosphate (3 $P_i$), short-chain (average ~ 7 $P_i$ units), and long-chain (average ~ 30 $P_i$ units) polyPs. Shown are dose–response curves with derived $IC_{50}$ estimates.

Arabidopsis and that PPK1-generated polyPs may force RcCHAD to relocate to the cytosol.

## Discussion

CHAD domains have been originally defined as "conserved histidine α-helical domains," with the histidines acting as metal chelators and/or phosphoacceptors (Iyer & Aravind, 2002). Although we found a $Zn^{2+}$ ion coordinated by two histidine residues in our RcCHAD structure (Fig S3), the contributing histidines are not conserved among CHAD family proteins (Fig S1), and no metal ions could be located in our CtCHAD and ygiF structures (Martinez et al, 2015). This makes it unlikely that CHAD domains are metal-binding proteins. Our structural analysis revealed that CHAD domains are helical bundles featuring an unusual internal symmetry. A set of

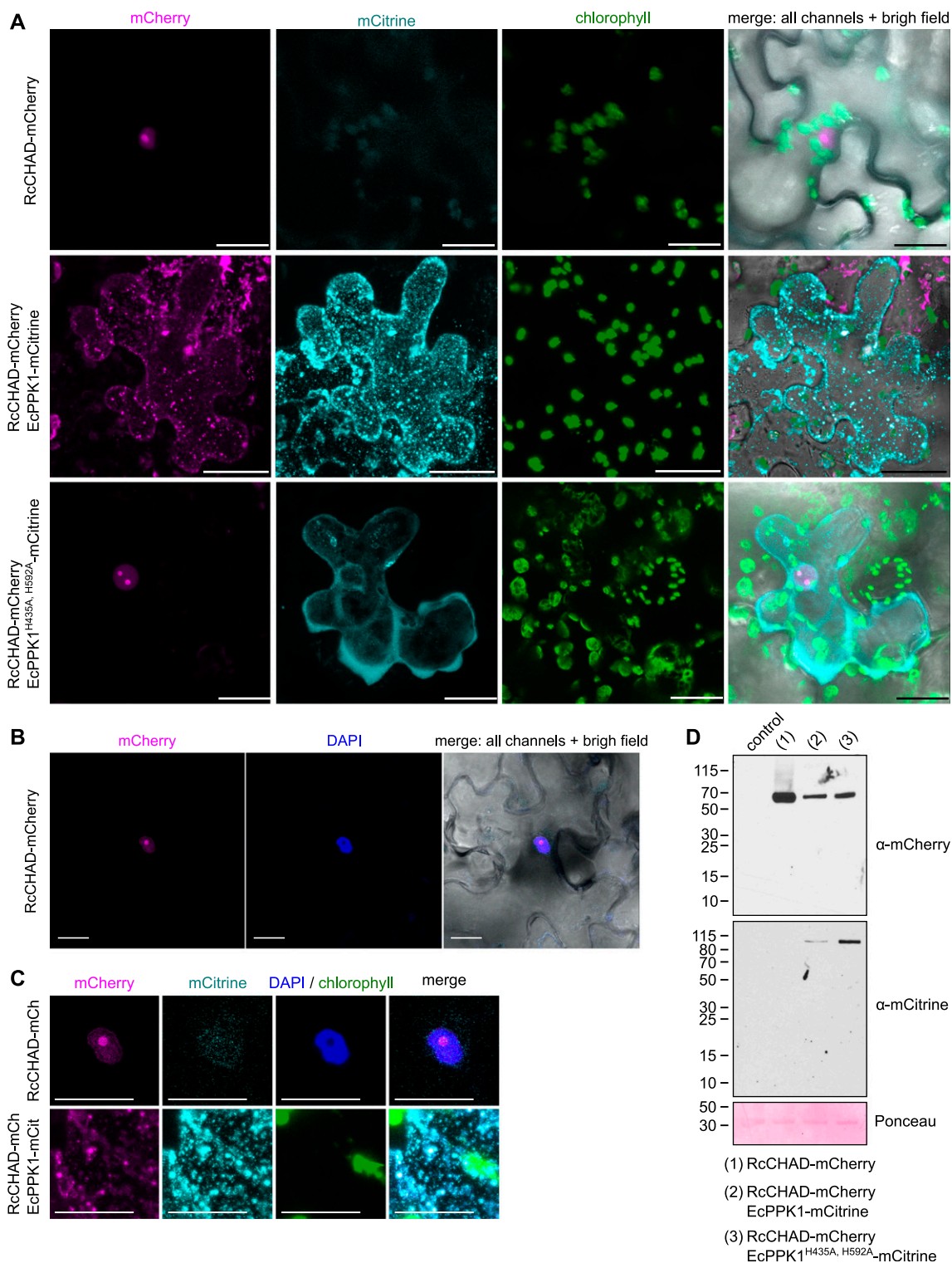

**Figure 6. RcCHAD localizes to the nucleus and nucleolus of tobacco cells and co-localizes with EcPPK1 to EcPPK1-generated polyP granules.**
**(A)** Transient expression of Ubi10p:RcCHAD-mCherry in tobacco leaves reveals a nuclear/nucleolar localization of the fusion protein (top row). Expression of Ubi10p: EcPPK1-mCitrine induces the formation of polyP granules (center row), not observed when using a catalytically impaired version of the enzyme (EcPPK1[H435A, H592A], bottom row). Scale bars correspond to 50 $\mu$m. Shown are Z-stacks from representative cells from three leaves obtained from three different plants. **(B)** RcCHAD-mCherry co-localizes with DAPI-stained nuclei and shows a higher intensity in nucleoli (not stained by DAPI). Scale bars correspond to 20 $\mu$m. **(C)** Magnified views of the nuclear localization of RcCHAD-mCherry when expressed in isolation (top row) and its redistribution to EcPPK1-generated polyP granules (bottom row). Scale bars correspond to 20 $\mu$m. **(D)** Western blots using anti-mCherry and anti-mCitrine antibodies reveal that RcCHAD-mCherry (63 kD) and EcPPK1-mCitrine (109 kD) migrated at the expected size in tobacco infiltrated leaves. RuBisCO (detected with Ponceau) is shown below as a loading control.

highly conserved basic amino acids contributes to the formation of a large basic surface area, which has evolved to sense long-chain polyPs but not nucleic acids or other $P_i$-containing ligands. Our plant, archaeal, and bacterial CHAD proteins bind polyPs with dissociation constants in the micromolar to nanomolar range, in good agreement with the cellular concentrations reported for polyPs in different organisms (Kumble & Kornberg, 1995; Bru et al, 2017; Rudat et al, 2018). Binding appeared specific, with the exception of heparin, which binds CtCHAD with high affinity but is absent in CHAD domain–containing organisms. It is of note that heparin also binds to the known polyP-specific dyes DAPI (Kolozsvari et al, 2014) and JC-D7/8 (Angelova et al, 2014). To our knowledge, heparin binding to the polyP-binding domain from *Escherichia coli* PPX1, which is used to detect polyP in immunofluorescence assays, has not been tested (Saito et al, 2005). Our quantitative biochemical experiments show that GCI can be used to quantify polyP binding to CHAD domains. In contrast to other methods, in GCI assays, the heterogeneous chain length of the polyP ligand does not affect the accuracy of the derived kinetic parameters. The sensograms of CtCHAD and ygiF binding to polyP could only be explained using a heterogeneous analyte model (Fig 3A). We speculate that different oligomeric states observed with the CHAD domains used in this study may account for this behavior (Fig 2D). Alternatively, it is possible that one CHAD domain may simultaneously bind to several polyP chains immobilized on the GCI chip. In line with this, size-exclusion chromatography coupled to right-angle light scattering (SEC-RALS) revealed the presence of CtCHAD dimers and tetramers in the absence of polyP (Fig S8). Addition of a long-chain polyP shifted CtCHAD into tetrameric and higher oligomeric states (Fig S8). It is of note that CtCHAD also shows a crystal packing consistent with a dimer or tetramer, which would enable cooperative binding of several CHAD domains to a single polyP chain (Fig S8). In any case, our competition assays and our structure-based polyP-binding site mutations in CtCHAD affected both binding kinetics, suggesting that our reported dissociation constants represent bona fide polyP-binding events (Fig 5B). Taken together, our binding assays and our polyP complex structure suggest that CHAD domains are polyP-binding modules that lack enzymatic activity (Martinez et al, 2015).

The conserved structural, biochemical, and sequence features of bacterial, archaeal, and eukaryotic CHAD domains suggest that these polyP-binding modules may have a common, ancient evolutionary origin. We identified an expressed CHAD domain–containing protein in the plant *R. communis* L. (RcCHAD), which represents the only CHAD domain currently known in plants. Interestingly, RcCHAD is ~80% sequence identical to a protein from the rhizosphere bacterium *Duganella* (Aranda et al, 2011) (Fig S1). Thus, *RcCHAD* might have been acquired by the plant via horizontal gene transfer from a soil-living bacterium.

Although we have biochemically characterized CHAD domains as polyP-binding proteins, their physiological roles remain to be defined. It has been previously shown that CHAD domain–containing proteins localize to polyP bodies in the bacterium *R. eutropha*, and that their over-expression can relocalize polyP granules to the cell poles (Tumlirsch & Jendrossek, 2017). However, genetic depletion of Ralstonia CHAD proteins or of ygiF did not result in any apparent phenotype (Kohn et al, 2012; Tumlirsch &

Jendrossek, 2017). Our enzymatic assays indicate that CHAD domains may assist polyP-metabolizing enzymes in recruiting their substrates. We speculate that the large polyP-binding surface in CHAD and its central cavity/pore (which is occupied by a polyP polymer in our complex structure, Fig 4B) may render fused polyP-metabolizing enzymes highly processive, as previously speculated (Alvarado et al, 2006). Consistently, about half of the annotated CHAD-containing proteins harbor N-terminal TTM domains, which we and others have previously characterized as short-chain inorganic polyphosphatases (Kohn et al, 2012; Moeder et al, 2013; Martinez et al, 2015).

In bacteria, polyP granules are spatially restricted in the nucleoid region (Racki et al, 2017). PolyP has also been shown to accumulate in the nucleolus of myeloma cells (Jimenez-Nuñez et al, 2012) and trypanosomes (Negreiros et al, 2018). We could observe a specific nuclear/nucleolar localization of RcCHAD in different cells and tissues when stably expressed in Arabidopsis, or transiently expressed in tobacco. We infer from this finding that polyP in plants may be located in the nucleolar compartment, as reported for animal cells (Jimenez-Nuñez et al, 2012). In line with this, ectopic expression of PPK1 leads to a relocalization of RcCHAD to presumed polyP granules in the cytosol. The polyP-binding domain from *E. coli* PPX1 has been previously used to detect polyP pools in fungal, trypanosomal, algal, and mammalian cells (Saito et al, 2005; Werner et al, 2007a, 2007b; Jimenez-Nuñez et al, 2012; Negreiros et al, 2018). Based on its small size, high polyP-binding affinity and specificity together with the well-characterized polyP-binding mechanism, we now propose CHAD domains as molecular probes to dissect polyP metabolism and storage in pro- and eukaryotic cells.

# Materials and Methods

## RT-PCR

RNA was extracted from ~100 mg of *R. communis* leaves with the RNeasy Plant Mini kit (QIAGEN). 2 μg of RNA was treated with DNase I (QIAGEN), copied to cDNA using an Oligo dT and SuperScript II Reverse Transcriptase (Invitrogen). RT-PCR was performed with primers RcCHAD_RT_F (5′-ATTGCCCAGGCAAAGCGTCATGC-3′) and RcCHAD_RT_R (5′-TTAGTGACGTAACTGTGGTGC-3′). The RT-PCR product was resolved on a 0.8% agarose gel, revealing a single DNA product. The band was excised and sequenced using the RcCHAD_RT_F/R primers. Sequencing results were analyzed using CLC Main Workbench 7.9.1 (QIAGEN).

## Generation of Arabidopsis transgenic lines

The *RcCHAD* coding sequence was cloned in pDONR221, the *UBIQUITIN10* promoter (pUBI10) in pDONR P4-P1R, and the mCherry fluorescence tag in pDONR P2R-P3, using the Gateway BP Clonase II Enzyme mix (Merck). The constructs were assembled with the Gateway LR Clonase enzyme mix (Merck) into the vector pH7m34GW (Karimi et al, 2005). *Agrobacterium tumefaciens* (pGV2260) was transformed with pH7m34GW harboring the construct pUBI10: RcCHAD-mCherry. *A. thaliana* was transformed using the floral

dip method (Clough & Bent, 1998), and plants were selected in ½ MS medium (½ MS [Duchefa], 1% [wt/vol] sucrose, 0.5 g/l MES, pH 5.7, 0.8% agar), supplemented with 20 μg/ml hygromycin.

## Transient protein expression in *Nicotiana benthamiana* leaves

The EcPPK1 coding sequence (UniProt ID C3T032) was cloned in pDONR221. PPK1 catalytic point mutations (H435A/H592A) were introduced by site-directed mutagenesis. EcPPK1 constructs were assembled together with pUBI10 (in pDONR P4-P1R) and mCitrine (in pDONR P2R-P3) into the pH7m34GW vector using the Gateway BP Clonase II Enzyme mix (Merck). *A. tumefaciens* was transformed with Ubi10p:RcCHAD-mCherry, Ubi10p:EcPPK1-mCitrine, Ubi10p:EcPPK1[H435A,H592A]-mCitrine, and p19. For each construct, 10 ml of *Agrobacterium* culture, grown overnight at 28°C, was collected by centrifugation. The cells were resuspended in 10 ml of infiltration solution (10 mM $MgCl_2$, 10 mM MES, pH 5.6, and 100 mM acetosyringon) and incubated for 3 h in darkness. For co-localization experiments, cells expressing two different constructs were mixed in equal volumes before infiltration. Cells transformed with p19 were added to all solutions. Tobacco leaves were infiltrated using a 0.5-ml syringe and plants were imaged after 2–5 d. Small pieces of leaves (~0.5 × 0.5 cm) were incubated for 1 h at room temperature with a PBS solution containing 5 μg/ml DAPI.

## Confocal microscopy

7-d-old *Arabidopsis* T3 seedlings expressing pUBI10:RcCHAD-mCherry, or tobacco leaves infiltrated with pUBI10:RcCHAD-mCherry and/or pUBI10:EcPPK1-mCitrine were imaged using an LSM 780 confocal microscope (Zeiss) equipped with a 40× NA 1.2 water C-Apochromat lens. Transmission was imaged at 514 nm. mCitrine and mCherry and DAPI were imaged using a GaAsP detector upon excitation at 514, 594, and 405 nm, respectively, and emission between 517 and 552 nm (mCitrine), 606-632 nm (mCherry), and 416–500 nm (DAPI), respectively. Chlorophyll was imaged with a photomultiplier tube detector upon excitation at 594 nm and with emission between 653 and 658 nm. Images were overlaid using Fiji (Schindelin et al, 2012).

## Western blotting

Arabidopsis seedlings or infiltrated tobacco leaves were snap-frozen in liquid nitrogen and homogenized with mortar and pestle. The plant extract was resuspended in 50 mM Tris, pH 8.0, 150 mM NaCl, 0.5% (vol/vol) Triton X-100, and cOmplete TM EDTA-free Protease Inhibitor Cocktail (Merck). 50 μg of protein extract (estimated by Bradford, Bio-Rad), pre-boiled for 5 min, was run on a 10% SDS–PAGE gel. Blotting was performed on a nitrocellulose membrane (GE Healthcare). After blocking with TBS buffer supplemented with 0.1% (vol/vol) Tween 20 and 5% (wt/vol) powder milk, the membrane was first incubated for 1 h with an anti-mCherry antibody (ab167453, dilution 1:2,000; Abcam), and then with an anti-rabbit peroxidase conjugate antibody (dilution 1:10,000, 1 h; Calbiochem). For mCitrine detection, the membrane was incubated for 1 h with an anti-GFP antibody coupled with HRP (Miltenyi Biotec) at 1:2,000 dilution. RuBisCO proteins were visualized with Ponceau

(0.1% [wt/vol] Ponceau S in 5% [vol/vol] acetic acid) as loading controls.

## Protein expression and purification

The coding sequences of RcCHAD (UniProt ID B9T8Q5), SsCHAD (UniProt ID Q97YW1), and CtCHAD[208–522] (UniProt ID Q8KE09) were obtained as synthetic genes from GeneArt (Life Technologies) and cloned into the vector pMH-HT (providing an N-terminal 6× His tag followed by a tobacco etch virus protease cleavage site) by Gibson assembly (Gibson et al, 2009) or restriction-based cloning. Plasmids were transformed in *E. coli* BL21 (DE3) RIL cells. For protein expression, the cells were grown in terrific broth medium at 37°C until $OD_{600}$ ~ 0.6, induced with 0.25 mM IPTG, and grown at 16°C for ~16 h. The cell pellets were collected by centrifugation at 4,500*g* for 30 min, resuspended in lysis buffer (50 mM sodium phosphate, pH 7.5, 500 mM NaCl, lysozyme, DNase I, and cOmplete Protease Inhibitor Cocktail [Merck]), and disrupted by sonication. The cell suspension was spun down at 18,000*g* for 1 h, and the supernatant was loaded onto an $Ni^{2+}$ affinity column (HisTrap HP 5 ml; GE Healthcare). The column was washed with 5 column volumes (CVs) of buffer A (50 mM PBS, pH 7.5, 500 mM NaCl), 5 CV buffer B (50 mM PBS, pH 7.5, 1 M NaCl), and 5 CV buffer C (250 mM PBS, pH 7.5, 500 mM NaCl). Proteins were eluted with buffer A supplemented with 0.5 M imidazole, pH 8.0, and cleaved overnight with tobacco etch virus at 4°C during dialysis in buffer A. RcCHAD was further purified by cation exchange (HiTrap SP HP cation exchange chromatography column; GE Healthcare), CtCHAD by size-exclusion chromatography, and SsCHAD by a second $Ni^{2+}$ affinity step. All samples were purified to homogeneity by size-exclusion chromatography on a Superdex 75 HR26/60 (GE Healthcare) equilibrated in buffer A. Protein concentrations were estimated by ultraviolet absorption at 280 nm ($A_{280\ nm}$) and using the respective theoretical molecular extinction coefficient calculated with the program PROTPARAM (https://web.expasy.org/protparam/). Mutations were introduced by site-directed mutagenesis, mutant proteins were purified like wild-type. Vtc4[189–487], ygiF-full length[1–433], ygiF-TTM[1–200], and ygiF-CHAD[201–433] were purified as described previously (Hothorn et al, 2009; Martinez et al, 2015).

## Crystallization

Hexagonal RcCHAD crystals developed at room temperature from hanging drops containing 1.5 μl of protein solution (1.6 mg/ml) and 1.5 μl of crystallization buffer (3 M NaCl, 0.1 M Bis-Tris, pH 6.0) suspended over 0.5 ml of crystallization buffer as reservoir solution. Orthorhombic CtCHAD crystals grew at room temperature in hanging drops containing 1.5 μl of protein (8 mg/ml) and 1.5 μl of crystallization buffer (2 M $(NH_4)_2SO_4$, 5% isopropanol). The CtCHAD—polyP complex was prepared by mixing CtCHAD at 8 mg/ml with short-chain polyP (BK Giulini GmbH, Calgon 188, average chain length ~7 $P_i$ units) to a final concentration of ~5 mM. Crystals developed in 0.4 M $(NH_4)_3PO_4$ in hanging drop setups. All crystals were cryoprotected by serial transfer into crystallization buffer supplemented with 20–30% ethylene glycol and snap-frozen in liquid nitrogen.

## Crystallographic data collection, structure solution, and refinement

Diffraction data were collected at beam line X06DA–PXIIIi of the Swiss Light Source, Villigen, Switzerland, and data processing and scaling was performed in XDS (version January 26, 2018) (Kabsch, 1993). For RcCHAD, a complete dataset at 2.3 Å resolution containing a weak anomalous signal to ~6 Å resolution (λ ~ 1.0 Å) was recorded (Table S1). A single $Zn^{2+}$ ion was located by SHELXD (Sheldrick, 2008) and the structure was solved using the single-wavelength anomalous dispersion method as implemented in the program phenix.autosol (Adams et al, 2010). The resulting model was completed in alternating cycles of manual model correction in the program COOT (Emsley & Cowtan, 2004) and restrained refinement in autoBUSTER (Global Phasing Ltd.). The sulfate ion-bound structure of CtCHAD was solved to 1.7 Å resolution using the molecular replacement method as implemented in PHASER (McCoy et al, 2007) (PDB-ID 3E0S was used as search model), and refined in phenix.refine (Adams et al, 2010). Isomorphous crystals of the CtCHAD—polyP complex diffracted to 2.1 Å resolution, restraints for a polyP 9-mer were generated using the program JLigand (Lebedev et al, 2012), and the structure was refined in REFMAC5 (Murshudov et al, 1997). Analysis with MolProbity revealed excellent stereochemisty for all refined models (Davis et al, 2007). Structural representations were done in Pymol (https://sourceforge.net/projects/pymol/) and using the ray tracer POVRAY (http://www.povray.org/). Secondary structure assignments were calculated with DSSP (Kabsch & Sander, 1983).

## Analytical size-exclusion chromatography

Gel filtration experiments were performed using a Superdex 200 Increase 10/300 GL column (GE Healthcare) pre-equilibrated in buffer A. 500 µl of the respective protein (0.5 mg/ml) was loaded sequentially onto the column, and elution at 0.75 ml/ml was monitored by ultraviolet absorbance at 280 nm. Peak fractions were analyzed by SDS–PAGE gel electrophoresis.

## Analysis of oligomeric states by means of RALS

The oligomeric states of "apo" and polyp-loaded CtCHAD were analyzed by SEC combined with RALS using an OMNISEC RESOLVE/REVEAL combo (Malvern), providing a GPC/SEC tetra-detector. Instrument constants were determined with defined concentrations of BSA. Apo CtCHAD, or CtCHAD incubated with 1 mM polyP (average chain length ~30 Pi units, BK Giulini GmbH, Calgon 322) at room temperature for 3 h, was analyzed in aliquots of 50 µl each at a sample concentration of 2 mg/ml (in 20 mM Hepes, pH 7.5, 150 mM NaCl) on a Superdex 200 10/300 increase column (GE Healthcare) at a column temperature of 35°C and a flow rate of 0.7 ml/min. The samples were analyzed using the OMNISEC software, version 10.41 (Fig S8).

## GCI binding assays

GCI assays were performed using a Creoptix WAVE system (Creoptix AG) (Kozma et al, 2009) as illustrated in Fig S6. Experiments were performed using 4PCP WAVE GCI chips (quasi-planar polycarboxylate surface; Creoptix AG). After conditioning with borate buffer (100 mM sodium borate pH 9.0, 1 M NaCl), the chip was immobilized in all channels with streptavidin and BSA via a standard amine-coupling: activation with 1:1 mix of 400 mM N-(3-dimethylaminopropyl)-N'-ethylcarbodiimide hydrochloride, and 100 mM N-hydroxysuccinimide, immobilization with 30 µg/ml of streptavidin in 10 mM sodium acetate, pH 5.0, passivation with 5% BSA in 10 mM sodium acetate, pH 5.0, and quenching with 1 M ethanolamine, pH 8.0. We did not succeed in coupling the CHAD domains directly to the chip using various methods. Hence, biotinylated medium chain polyP (5–20 µg/µl; Kerafast), 5'-biotinylated single-strand DNA (5 µg/µl, 54 nucleotides; Metabion), 5'-biotinylated RNA (5 µg/µl, polyG. 10 nucleotides; Microsynth) or biotinylated heparin (Merck) was bound to the chip surface. Analytes were injected in a 1:2 dilution series in 50 mM Bis-Tris, pH 7.5, 150 mM NaCl at 25°C. Blank injections (every three cycles) were used for double referencing and a DMSO calibration curve (0–2% DMSO, four dilutions) for bulk correction. Data analysis was performed using the Creoptix WAVEcontrol software version 3.5.13 (applied corrections: X and Y offset, DMSO calibration, double referencing), and a one-to-one binding model was used to fit all experiments with the exception of CtCHAD and ygiF, in which we used a heterogeneous analyte model. For competition experiments, fixed concentration of CHAD proteins were incubated with a dilution series of sodium tripolyphosphate (Merck), short-chain polyP (average chain length ~7 $P_i$ units, BK Giulini GmbH, Calgon 188), polyP (average chain length ~30 $P_i$ units, BK Giulini GmbH, Calgon 322), or $P^1$, $P^5$ -Di(adenosine-5') pentaphosphate pentasodium salt (Merck).

## Phosphohydrolase activity measurements

10 nM of ygiF full-length1-433 and ygiF-TTM1-200 were incubated for 7 min at 37°C with 500 µM of substrate in reaction buffer (20 mm Bis-Tris propane, pH 8.5, 150 mm NaCl, and 5 mm MgCl2). The substrates tested were ATP (Merck), sodium tripolyphosphate (Merck), and polyP (average length ~30 Pi units, BK Giulini GmbH, Calgon 322). 100 µl of the reaction was incubated for 5 min with 28 µl of a malachite green solution containing 3 mM malachite green, 15% (vol/vol) sulfuric acid, 1.5% molybdate (wt/vol), and 0.2% (vol/vol) Tween 20. The absorption at A595 nm was measured using a synergy H4 plate reader (BioTek). Blanks were obtained for each substrate by adding heat-inactivated enzyme (boiled for 5 min at 95°C) to the respective reactions. Experiments were performed in triplicates.

# Supplementary Information

# Acknowledgements

We thank the staff at beam-line X06DA—PXIII of the Swiss Light Source, Villigen, Switzerland, for their technical help during data collection and F Spiga for advice on the GCI experiments and critical reading of the

manuscript. This work was supported by an ERC starting grant from the European Research Council under the European Union's Seventh Framework Programme (FP/2007-2013)/ERC Grant Agreement no 310856 and the Howard Hughes Medical Institute (International Research Scholar Award; both to M Hothorn)

## Author Contributions

L Lorenzo-Orts: conceptualization, data curation, formal analysis, validation, investigation, visualization, methodology, and writing—original draft, review, and editing.
U Hohmann: methodology and writing—review and editing.
J Zhu: methodology and writing—review and editing.
M Hothorn: conceptualization, resources, data curation, formal analysis, supervision, funding acquisition, validation, investigation, visualization, methodology, project administration, and writing—original draft, review, and editing.

## Conflict of Interest Statement

The authors declare that they have no conflict of interest.

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
