## [Reviewer comments · Life Science Alliance]

Life Science Alliance

Molecular characterization of CHAD domains as inorganic polyphosphate binding modules

Laura Lorenzo-Orts, Ulrich Hohmann, Jinsheng Zhu, and Michael Hothorn

DOI: <https://doi.org/10.26508/lsa.201900385>

Corresponding author(s): Michael Hothorn, University of Geneva

Review Timeline:

Submission Date:	2019-03-20
Editorial Decision:	2019-04-13
Revision Received:	2019-05-16
Editorial Decision:	2019-05-20
Revision Received:	2019-05-20
Accepted:	2019-05-20

Scientific Editor: Andrea Leibfried

Transaction Report:

April 13, 2019

Re: Life Science Alliance manuscript #LSA-2019-00385-T

Prof. Michael Hothorn
University of Geneva
Department of Botany and Plant Biology
Science III
30 Quai E. Ansermet
Geneva CH-1211
Switzerland

Dear Dr. Hothorn,

Thank you for submitting your manuscript entitled "Molecular characterization of CHAD domains as inorganic polyphosphate binding modules" to Life Science Alliance. The manuscript was assessed by expert reviewers, whose comments are appended to this letter.

As you will see, the reviewers appreciate the high quality of your work and provide constructive input on how to further strengthen it. We would thus like to invite you to submit a revised version to us, addressing the individual points raised by the reviewers. Please assess the effects of RNA and heparin (reviewer #1). Please also provide better insight into the potential physiological significance in planta (reviewer #3).

Thank you for this interesting contribution to Life Science Alliance. We are looking forward to receiving your revised manuscript.

Sincerely,

B. MANUSCRIPT ORGANIZATION AND FORMATTING:

Reviewer #1 (Comments to the Authors (Required)):

This manuscript describes a careful and comprehensive study on the structure and function of CHAD domains. These protein domains had previously no assigned function. The authors

demonstrated convincingly that these domains bind the polyanion polyP. The authors solved the structure of various CHAD domains, and conducted careful mutagenesis studies to define the client specificity. Finally, the authors expressed CHAP-mcherry fusions in Arabidopsis to illustrate the potential application of this fusion to detect polyP *in vivo*.

I have only a few minor and one slightly more major comment that need to be addressed:

1) the authors might want to consider using one of their purified CHAD domains (ideally the one with the highest affinity for polyP) and conduct affinity measurements with RNA and heparin. Both are frequently interfering with polyP detection *in vivo*. This becomes particularly important since the authors see the CHAD-domain to form foci in the nucleus/nucleolus.

2) It is not completely obvious from the graph why the authors speculate that RcCHAD also behaves like an oligomer. There is some impurity in the protein- could this explain the appearance of 2 peaks?

3) Fig. 3B/5E - the authors might want to explain why their y-axis range is so different

4) label "B" in fig. 5B should be above the first panel on the right.

5) there are several issues with the IF data:

- it is unclear why the signal of the CHAD-mcherry is so much higher in the PPK expressing cells than in the cells that either do not express PPK or express the mutant? Wouldn't one expect a much brighter signal in those cells which gets more diffuse as polyP accumulates in the cytosol? Is it possible that the expression of active PPK affects the expression level of CHAD-mcherry?

Westernblot analysis should be presented to show that all cells express the same level of CHAD-mcherry and, in the respective cells, the same amount of wildtype and mutant PPK.

- nuclear marker need to be included

- the right and left panels seem not to be in focus

Reviewer #2 (Comments to the Authors (Required)):

The authors set out to examine the structure of the CHAD domains from bacterial, archaeal, and eukaryotic origin and report that they are polyphosphate binding domains. The authors found dissociation constants ranging from the nano- to mid-micromolar range. Mutational analyses of CHAD-polyP validated the complex structure. They localized the domain to the nucleus and nucleolus of plant cells. This is a thoughtfully considered and well executed study. Overall, this manuscript advances the field significantly as it stands. I have only minor suggestions.

Line 43: blood coagulation

Line 189 (and other places): amino acids (separated)

Line 229: polyphosphate is also present in the nucleolus of trypanosomes (Negreiros et al., Mol. Microbiol. 110 973-996, 2018).

Line 235: the polyP binding domain was also used to localize polyP in trypanosomes (same reference above, Negreiros et al., 2018)

Line 482: Mutations of con?

Line 543: passivated?

Reviewer #3 (Comments to the Authors (Required)):

In the work the authors undertook the characterization of the conserved CHAD domain present in

all three kingdoms of life. They convincingly demonstrate that the CHAD is a polyP binding domain that does not seem to have an intrinsic polyP hydrolyzing activity, but may enhance the enzymatic activity of polyphosphatases (ygiF) or may be involved in localizing polyP to the nucleus and the nucleolus in plant cells. The work seems technically sound and of high standard using state of the art technology.

Hence, my comments relate to the general significance of the work.

1) The hypothesis of a role of the CHAD domain in polyP metabolism/localization in plants is interesting but the authors do not show any evidence that polyphosphates indeed exist in plants. Indeed, the CHAD domain is often associated with proteins containing a TTM domain, which in some cases (bacteria for instance) is a enzymatically competent polyphosphatase. However, plant (*A. thaliana*) TTM3 is a tripolyphosphatase and not a long chain polyphosphatase. Could the CHAD domain also bind short chain poly(tri)phosphates ?

2) Another point is that, according to my understanding of the manuscript, the only higher plant where a CHAD domain has been identified is *Ricinus communis*. The CHAD domain seems to be absent in *A. thaliana* for instance. This suggests that, at least in plants, the CHAD domain is of limited importance and its presence in *R. communis* may be linked to some peculiarities of *Ricinus* secondary metabolism. Could polyphosphates have a specific role in this species?

For the reader not familiar with plant biology (such as this reviewer) it would be interesting to discuss these possibilities. In addition, and in view of the above arguments (mainly absence of proof for the existence of polyP in plants, CHAD documented in only one plant species), I feel that the main conclusion of the manuscript (i.e. last four line of the abstract) is somewhat overinterpreted as far as plants are concerned, but may be correct in prokaryotes.

1. We have reformatted the manuscript to meet LSA style (figure labels and sizes, reference format)
2. We have shortened the abstract to 173 words.
3. We have added a summary blurb (lines 9-11) that reads: “**A domain of unknown function termed CHAD, present in all kingdoms of life, is characterized as a specific inorganic polyphosphate binding domain.**”
4. We have added a conflict of interest statement.
5. We have renamed the methods section to meet LSA style and reordered the manuscript according to your formatting suggestions.
6. Data availability: The crystallographic structure factors and coordinates have been deposited with the Protein Data Bank. The stable RcCHAD-mCherry transgenic line will be deposited with the NASC stock center. Expression constructs will be deposited with Addgene. Full blots and gels are shown in figures 1D, 2D, 3E, 6D. For the biochemical GCI binding assays, raw traces are presented in figures 3, 5 and S7. All other materials including the raw confocal images have been compiled and will be available upon request.

Response to Reviewer #1:

This manuscript describes a careful and comprehensive study on the structure and function of CHAD domains. These protein domains had previously no assigned function. The authors demonstrated convincingly that these domains bind the polyanion polyP. The authors solved the structure of various CHAD domains, and conducted careful mutagenesis studies to define the client specificity. Finally, the

authors expressed CHAP-mcherry fusions in *Arabidopsis* to illustrate the potential application of this fusion to detect polyP *in vivo*.

I have only a few minor and one slightly more major comment that need to be addressed:

1) the authors might want to consider using one of their purified CHAD domains (ideally the one with the highest affinity for polyP) and conduct affinity measurements with RNA and heparin. Both are frequently interfering with polyP detection *in vivo*. This becomes particularly important since the authors see the CHAD-domain to form foci in the nucleus/nucleolus.

OUR RESPONSE:

We thank reviewer #1 for her/his suggestion. We further tested the ligand specificity of CtCHAD against biotinylated RNA (a polyG 10mer) and biotinylated heparin, and using biotinylated polyP as a control in quantitative GCI assays. The results are shown in revised Figure 3D and are described in the revised text (lines 136-139): “...and we could not observe detectable binding of SsCHAD or RcCHAD to GCI chips coated with biotinylated single-stranded DNA or RNA (Fig. 3D). However, heparin, which is absent in bacteria and plants, was bound by CtCHAD with binding constants comparable to polyP (Fig 3D).”

We have added a statement in our discussion that highlights the fact that heparin binds to CHAD domains and also to the known polyP-’specific’ dyes DAPI and JC-D7/8 (lines 208-213): “Binding appeared specific, with the exception of heparin, which binds CtCHAD with high affinity but is absent in CHAD domain-containing organisms. It is of note that heparin also binds to the known polyP-specific dyes DAPI (Kolozsvari *et al*, 2014) and JC-D7/8 (Angelova *et al*, 2014). To our knowledge, heparin binding to the polyP-binding domain from *E. coli* PPX1, which is used to detect polyP in immunofluorescence assays, has not been tested (Saito *et al*, 2005).“

2) It is not completely obvious from the graph why the authors speculate that RcCHAD also behaves like an oligomer. There is some impurity in the protein- could this explain the appearance of 2 peaks?

OUR RESPONSE:

Indeed there is an impurity in all our RcCHAD preparations, which appears to co-purify with the plant CHAD domain expressed in *E. coli*. We found the RcCHAD protein contained in both peaks shown in the analytical size exclusion chromatography experiments in Figure 2D. We have purified RcCHAD again to performed SEC-RALS analysis (as shown for the well-behaving CtCHAD in Fig. S8). However, the protein degrades during the run at the required column temperature of 35 deg C. We thus have removed the statement referring to the oligomeric state of RcCHAD from the manuscript. The

revised section now reads (lines 107-108): "...with ygiF and SsCHAD behaving as monomers, while unliganded CtCHAD form dimers in solution (Figs 2D and S8, see Discussion)."

3) Fig. 3B/5E - the authors might want to explain why their y-axis range is so different

OUR RESPONSE

These apparent discrepancies can be explained by the different CHAD proteins being used in the assays presented in Figures 3B versus 5E. The observed differences in surface mass originate from different protein concentrations used as analytes in the different assays. These differences do not affect the accuracy of the IC50 values derived from these experiments.

4) label "B" in fig. 5B should be above the first panel on the right.

OUR RESPONSE

Corrected.

5) there are several issues with the IF data:

OUR RESPONSE:

We apologize for not having explained this experiment better in our manuscript. The data shown in Fig. 6 represent live imaging confocal microscopy of fluorescently tagged RcCHAD and EcPPK1 proteins expressed in *Nicotiana benthamiana* leaves.

- it is unclear why the signal of the CHAD-mcherry is so much higher in the PPK expressing cells than in the cells that either do not express PPK or express the mutant? Wouldn't one expect a much brighter signal in those cells which gets more diffuse as polyP accumulates in the cytosol?

OUR RESPONSE:

These experiments were done by transiently expressing the respective fusion proteins in tobacco. In this system expression varies greatly from cell to cell. Thus this method can be used to study the sub-cellular localization of proteins, but not to quantify or to compare protein levels between different experiments (Bashandy et al., Plant Methods 2015). We found nuclear/nucleolar localization of RcCHAD-mCherry in different wild-type and EcPPK1^{H435A,H592A}-mCitrine expressing cells, but a cytoplasmic localization of RcCHAD-mCherry in active, polyP-generating EcPPK1 cells (see recent method paper for comparison: <https://www.biorxiv.org/content/10.1101/630129v2>). To better visualize the different localization patterns, we have included Z-stacks in the revised Figure 6. The

corresponding figure legend now reads: “Shown are Z-stacks from representative cells from three leaves obtained from three different plants.”

Is it possible that the expression of active PPK1 affects the expression level of CHAD-mcherry? Western blot analysis should be presented to show that all cells express the same level of CHAD-mcherry and, in the respective cells, the same amount of wild-type and mutant PPK.

OUR RESPONSE

We performed western blot analysis of tobacco leaves infiltrated with (i) *RcCHAD* alone, (ii) *RcCHAD* and *EcPPK1*, and (iii) *RcCHAD* and *EcPPK1* mutant (revised Figure 6D). We detect single bands migrating at the predicted molecular weights of *RcCHAD* and *EcPPK1*. Hence, the fluorescent signals correspond to full-length *RcCHAD*-mCherry and *EcPPK1*-mCitrine, and are not derived from free mCherry or free mCitrine, respectively.

We observe that *RcCHAD* expression is higher in leaves infiltrated with only *RcCHAD* than in leaves co-infiltrated with *RcCHAD* and *EcPPK1*. We cannot discard that *EcPPK1* expression might affect *RcCHAD* expression. Nevertheless, the aim of this experiment was to examine if *RcCHAD* co-localizes with PPK1-derived granules, not to compare proteins levels, for which this experiment is not suited (see above). The localization of *RcCHAD* does not depend on its protein levels, since cells from different leaves infiltrated with *RcCHAD* showed different fluorescent intensity.

In order to control proteins levels, we would need to generate single transgenic Arabidopsis lines for the different proteins, select lines with comparable protein levels, and then obtain genetic crosses between them. This is a very time-consuming experiment, which is further complicated by the fact that PPK1 expression in Arabidopsis is highly toxic (<https://www.biorxiv.org/content/10.1101/630129v1>).

- nuclear marker need to be included

OUR RESPONSE

We have performed the suggested control experiment, which is shown in revised Figure 6B. We used DAPI, which stains the nucleus of plant cells (see for instance Gallemí et al., Development 2016) but not the nucleolus (Sirri et al., Histochem Cell Biol. 2008). We find that *RcCHAD*-mCherry co-localizes with DAPI in the nucleus of tobacco cells, and shows a higher signal in the nucleolus (which is not stained by DAPI). When *RcCHAD* is co-expressed with *EcPPK1*, both proteins co-localize to cytosolic granules which do not correspond to chloroplasts (as judged from the chlorophyll autofluorescence) (Fig. 6A,C). Thus, DAPI staining confirms the nuclear localization of isolated *RcCHAD*-mCherry.

- the right and left panels seem not to be in focus

OUR RESPONSE

We have included Z-stacks in our revised Fig. 6A to account for this.

Reviewer #2:

The authors set out to examine the structure of the CHAD domains from bacterial, archaeal, and eukaryotic origin and report that they are polyphosphate binding domains. The authors found dissociation constants ranging from the nano- to mid-micromolar range. Mutational analyses of CHAD-polyP validated the complex structure. They localized the domain to the nucleus and nucleolus of plant cells. This is a thoughtfully considered and well executed study. Overall, this manuscript advances the field significantly as it stands. I have only minor suggestions.

Line 43: blood coagulation Line 189 (and other places): amino acids (separated)

OUR RESPONSE

Corrected throughout.

Line 229: polyphosphate is also present in the nucleolus of trypanosomes (Negreiros et al., Mol. Microbiol. 110 973-996, 2018).

OUR RESPONSE

Statement and reference added.

Line 235: the polyP binding domain was also used to localize polyP in trypanosomes (same reference above, Negreiros et al., 2018)

OUR RESPONSE

Statement and reference added.

Line 482: Mutations of con?

OUR RESPONSE

Corrected.

Line 543: passivated?

OUR RESPONSE

This is the correct term.

Reviewer #3:

In the work the authors undertook the characterization of the conserved CHAD domain present in all three kingdoms of life. They convincingly demonstrate that the CHAD is a polyP binding domain that does not seem to have an intrinsic polyP hydrolyzing activity, but may enhance the enzymatic activity of polyphosphatases (ygiF) or may be involved in localizing polyP to the nucleus and the nucleolus in plant cells. The work seems technically sound and of high standard using state of the art technology.

Hence, my comments relate to the general significance of the work.

1) The hypothesis of a role of the CHAD domain in polyP metabolism/localization in plants is interesting, but the authors do not show any evidence that polyphosphates indeed exist in plants.

OUR RESPONSE

There are a number of early studies that report polyP stores in higher plants (for example Miyachi (1961) J Biochem, 50:4-367-371). We have recently used the polyP-specific dye JC-D7 and the polyP-binding domain from EcPPX1 to localize polyPs in plant cells and tissues, but only found significant stores in the green algae *Chlamydomonas*. From this, we conclude that there are no highly concentrated polyP granules in higher plants, at least not under the growth conditions and tissues tested (<https://www.biorxiv.org/content/10.1101/630129v1>).

The accumulation of RcCHAD in the nucleus/nucleolus region of plant cells however suggests that polyP might exist in these cellular compartments. This hypothesis needs to be tested in future studies. We have thus toned down our abstract accordingly, which now reads (lines 22-25): “The presence of a CHAD domain in the polyPase ygiF enhances its enzymatic activity. **The only known CHAD protein from the plant *Ricinus communis* localizes to the nucleus/nucleolus when expressed in Arabidopsis and tobacco, suggesting that plants may harbor polyPs in these compartments.** We propose that CHAD domains may be used to engineer the properties of polyP-metabolizing enzymes and to specifically localize polyP stores in eukaryotic cells and tissues.”

Indeed, the CHAD domain is often associated with proteins containing a TTM domain, which in some cases (bacteria for instance) is a enzymatically competent polyphosphatase. However, plant (*A. thaliana*) TTM3 is a tripolyphosphatase and not a long chain polyphosphatase. Could the CHAD domain also bind short chain poly(tri)phosphates?

OUR RESPONSE

We analyzed binding of CtCHAD to tripolyphosphate in competition assays. We observe that CtCHAD also binds tripolyphosphate, showing a similar IC50 that the one obtained for a polyP 7mer (see revised Figure 5E).

2) Another point is that, according to my understanding of the manuscript, the only higher plant where a CHAD domain has been identified is *Ricinus communis*. The CHAD domain seems to be absent in *A. thaliana* for instance. This suggests that, at least in plants, the CHAD domain is of limited importance and its presence in *R. communis* may be linked to some peculiarities of *Ricinus* secondary metabolism. Could polyphosphates have a specific role in this species? For the reader not familiar with plant biology (such as this reviewer) it would be interesting to discuss these possibilities.

OUR RESPONSE

Indeed, we speculate that RcCHAD might have been acquired by *Ricinus* via horizontal gene transfer from bacteria. Genes acquired in this way can either maintain the same function in the host organism or adopt a new one. We could thus far not detect polyP granules in *Ricinus* (shown in Figure S4 in our method manuscript <https://www.biorxiv.org/content/10.1101/630129v2>). However, *Ricinus communis* is used in phytoremediation of toxic metals, as for instance copper (Bauddh et al., 2015). Interestingly, *Ricinus* capacity to bind copper increases when plants are grown under phosphate surplus (Huang et al., Environ Sci Pollut Res Int. 2018). In *E. coli*, copper tolerance has been linked to polyP degradation, and cells grown in high phosphate media are thereby more resistant to copper toxicity (Grillo-Puertas et al., BMC Microbiol. 2014). Hence, a potential role of polyP in *Ricinus* could be in protecting the plant from metal toxicity. While we would prefer not to include such speculative statements in our current manuscript, we have currently a review under consideration at *New Phytologist*, in which the potential roles of RcCHAD and polyPs in plants are critically discussed.

In addition, and in view of the above arguments (mainly absence of proof for the existence of polyP in plants, CHAD documented in only one plant species), I feel that the main conclusion of the manuscript (i.e. last four line of the abstract) is somewhat overinterpreted as far as plants are concerned, but may be correct in prokaryotes.

OUR RESPONSE

We fully agree with reviewer #3' assessment. We have revised the respective statement in our abstract (lines 22-26), which now reads: “**The only known CHAD protein from the plant *Ricinus communis* localizes to the nucleus/nucleolus when expressed in Arabidopsis and tobacco, suggesting that plants may harbor polyPs in these compartments.** We propose that CHAD domains may be used to engineer

the properties of polyP-metabolizing enzymes and to specifically localize polyP stores in eukaryotic cells and tissues.”

May 20, 2019

RE: Life Science Alliance Manuscript #LSA-2019-00385-TR

Prof. Michael Hothorn
University of Geneva
Department of Botany and Plant Biology
Science III
30 Quai E. Ansermet
Geneva CH-1211
Switzerland

Dear Dr. Hothorn,

Thank you for submitting your revised manuscript entitled "Molecular characterization of CHAD domains as inorganic polyphosphate binding modules". As you will see, the reviewers appreciate your point-by-point response and the introduced changes and we would thus be happy to publish your paper in Life Science Alliance. Please log into our system one more time to fill in the electronic license to publish form. Your new manuscript number will be LSA-2019-00385-TRR, please make sure to carry over all manuscript files to this new number (single click process).

A. FINAL FILES:

B. MANUSCRIPT ORGANIZATION AND FORMATTING:

Sincerely,

Reviewer #1 (Comments to the Authors (Required)):

The authors have adequately addressed my concerns.

Reviewer #2 (Comments to the Authors (Required)):

I am satisfied with the responses and modifications to the manuscript

Reviewer #3 (Comments to the Authors (Required)):

The authors have addressed all the issues raised.

May 20, 2019

RE: Life Science Alliance Manuscript #LSA-2019-00385-TRR

Prof. Michael Hothorn
University of Geneva
Department of Botany and Plant Biology
Science III
30 Quai E. Ansermet
Geneva CH-1211
Switzerland

Dear Dr. Hothorn,

Thank you for submitting your Deleted entitled "Molecular characterization of CHAD domains as inorganic polyphosphate binding modules". It is a pleasure to let you know that your manuscript is now accepted for publication in Life Science Alliance. Congratulations on this interesting work.

DISTRIBUTION OF MATERIALS:

Again, congratulations on a very nice paper. I hope you found the review process to be constructive and are pleased with how the manuscript was handled editorially. We look forward to future exciting submissions from your lab.

Sincerely,

Andrea Leibfried, PhD
Executive Editor
Life Science Alliance
Meyerohofstr. 1
69117 Heidelberg, Germany
t +49 6221 8891 502
e a.leibfried@life-science-alliance.org
www.life-science-alliance.org